# Protective Mechanism of *Eurotium amstelodami* from Fuzhuan Brick Tea against Colitis and Gut-Derived Liver Injury Induced by Dextran Sulfate Sodium in C57BL/6 Mice

**DOI:** 10.3390/nu16081178

**Published:** 2024-04-16

**Authors:** Xin Wang, Jinhu Liu, Jianping Wei, Yuxiang Zhang, Yunpeng Xu, Tianli Yue, Yahong Yuan

**Affiliations:** 1College of Health Management, Shangluo University, Shangluo 726000, China; 202226@slxy.edu.cn (X.W.); 202228@slxy.edu.cn (J.L.); 2Shaanxi Union Research Center of University and Enterprise for Healthy and Wellness Industry, Shangluo 726000, China; 3College of Food Science and Technology, Northwest University, Xi’an 710069, China; jianpingwei0327@nwu.edu.cn (J.W.); zhangyx729@nwu.edu.cn (Y.Z.); yuetl305@nwafu.edu.cn (T.Y.); 4Shangluo Characteristic Industry and Leisure Agriculture Guidance Center, Shangluo 726000, China; wangxinnwsuafedu@163.com

**Keywords:** *Eurotium amstelodami*, inflammatory cytokines, intestinal microflora, metabolome, ulcerative colitis

## Abstract

The study explored the potential protective impact of the probiotic fungus *Eurotium amstelodami* in Fuzhuan brick tea on ulcerative colitis, along with the underlying mechanism. A spore suspension of *E. amstelodami* was administered to C57BL/6 mice to alleviate DSS-induced colitis. The findings indicated that administering *E. amstelodami* evidently enhanced the ultrastructure of colonic epithelium, showing characteristics such as enhanced TJ length, reduced microvilli damage, and enlarged intercellular space. After HLL supplementation, the activation of the liver inflammation pathway, including TLR4/NF-kB and NLRP3 inflammasome caused by DSS, was significantly suppressed, and bile acid metabolism, linking liver and gut, was enhanced, manifested by restoration of bile acid receptor (FXR, TGR5) level. The dysbiosis of the gut microbes in colitis mice was also restored by HLL intervention, characterized by the enrichment of beneficial bacteria (*Lactobacillus*, *Bifidobacterium*, *Akkermansia*, and *Faecalibaculum*) and fungi (*Aspergillus*, *Trichoderma*, *Wallemia*, *Eurotium*, and *Cladosporium*), which was closely associated with lipid metabolism and amino acid metabolism, and was negatively correlated with inflammatory gene expression. Hence, the recovery of gut microbial community structure, implicated deeply in the inflammatory index and metabolites profile, might play a crucial role in the therapeutic mechanism of HLL on colitis.

## 1. Introduction

Inflammatory bowel disease (IBD), encompassing conditions like Crohn’s disease and ulcerative colitis, has evolved into a global health concern [1], characterized by persistent and recurrent inflammation of the gastrointestinal tract. Although the precise pathogenesis of IBD remains elusive, various studies have proposed a link between genetic and environmental factors and chronic intestinal inflammation. The regression of inflammation promotes the recovery of proper tissue repair [2]. Current experimental evidence has shown that several pro-inflammatory cytokines, including TNF-α, IL-6, and IL-1β, are upregulated during colitis progression, linked closely to the activation of the NF-κB pathway [3]. In addition, mononuclear hematopoietic and intestinal epithelial cells serve as the first line for colon defense and mediate the innate immune response stimulated by microbes, nucleotide-binding oligomerization domain protein-like receptors (NLRs), or inflammasome proteins such as the NLR family pyrin domain (NLRP1, NLRP3, and NLRC4) [4]; among these, the NLRP3 inflammasome was recognized as a crucial mechanism in the DSS colitis model, and the imperative control of inflammatory response involves the negative regulation of NLRP3 activity [5]. Notably, under the stimulation of external factors, endogenous transcription factors such as NF-κB can drive the expression of inflammatory molecules, including the components of NLRP3. Therefore, the suppression of the NF-κB and NLRP3 pathway may be an effective target to reduce colitis. Moreover, the focus extended to the influence of intestinal inflammation on liver health. The gut–liver axis relies on the integrity of the intestinal barrier for maintaining liver homeostasis [6]. The liver serves as a secondary defense against potentially harmful substances translocated from the gut and is also involved in regulating the mucosal barrier [7]. The gut–liver axis establishes a connection between the liver and the intestine through bile acid metabolism, and the regulatory roles of bile acids are primarily mediated by receptors, including the nuclear receptors farnesoid-X receptor (FXR) and pregnane-X-receptor (PXR), as well as the membrane G-protein-coupled receptor (GPCR; TGR5) [8,9].

Gut environments, comprising the microbiota and their metabolites, impact the preservation of gut homeostasis. Importantly, addressing gut microbiota dysbiosis is a key focus for developing novel therapies in experimental colitis [10]. Mounting evidence suggests that the gut microbiome significantly contributes to the pathology and progression of IBD due to its close interaction with the host immune system [11]. The gut microbiota, acknowledged as an essential ‘metabolic organ’, performs vital functions in preserving human health and triggering diseases. These functions encompass digestion, nutrient absorption, energy supply, immune regulation, disease resistance, etc. [12,13]. The disruptions of gut microbiome and their impact on metabolic and physiological functions exert an indispensable role as well [14]. Research indicates that gut dysbiosis can disrupt both host immune responses and gut barrier function [15]. Thus, the modulation of gut microbiota as a potential therapeutic target for UC is under exploration. Furthermore, the metabolic activity stands out as a crucial characteristic of the gut microbiota, representing one of the key mechanisms in the interaction between the host and microbiota [16]. These microorganisms possess a diverse array of enzymes, enabling the metabolism of various substrates, including beneficial substances and pro-inflammatory cytokines [17]. Meanwhile, metabolomic analysis is gaining prominence as a potent method in systems biology research, offering distinctive insights into understanding organisms, diagnosing diseases, exploring pathology, and investigating toxicology. For example, metabolomics analyses of serum samples from CD and UC patients have revealed that most metabolites were annotated as the phospholipids, which are significantly reduced relative to the levels in healthy individuals.

Inspired by these findings, the introduction of targeted probiotics can alleviate inflammation symptoms and support the establishment of a well-balanced intestinal microbial structure and metabolic activities [18]. The manipulation of gut microbiota through probiotic administration offers promising therapeutic options for IBD. It has long been suggested that incorporating potential probiotics through fermented products can enhance human health [19]. Fermentation is a valuable process in food preservation, while also enhancing flavor and improving nutritional properties. Examples include pickles, sauerkraut, sourdough bread, kefir, craft beers, and kombucha (fermented tea). This ancient practice, dating back to the seventh millennium BC, involves the presence of various bacteria and fungi [20]. Notably, Fuzhuan brick tea (FBT) is fermented in the presence of one predominant species, *Eurotium amstelodami*, commonly referred to as the ‘golden flora’ in Chinese due to its yellow cleistothecia [21,22]. Evidence from a previous study showed that FBT consumption has the potential to ameliorate metabolic disease conditions [23]. Since the dominant fungus *E. amstelodami* in FBT significantly contributes to both its flavor and color, we thus hypothesized that *E. amstelodami* could survive in the gastrointestinal tract and produce effects in our study.

To further explore the potential protective mechanism for *E. amstelodami* against colitis mice, we used a dextran sulfate sodium (DSS)-induced colitis model. We further conducted a comprehensive study on the effects of *E. amstelodami* on the gut microbiota of C57BL/6 mice via combining fecal metabonomics and 16S&ITS rRNA gene sequencing analysis from the perspective of intestinal microbiota.

## 2. Materials and Methods

### 2.1. Preparation of Spore Suspension

*E. amstelodami* H-1 and *E. amstelodami* S-6 were separately isolated from marketed FBT (Hunan Baishachong Tea Co., Ltd., Yiyang, China; Shaanxi Daqin Tea Idustry Co., Ltd., Xi’an, China) in our laboratory, were cultured on potato dextrose agar (PDA), and were stored at 4 °C. Two strains were cultured in M40Y medium at 28 °C for 48 h to harvest their spores. The spore concentration was then adjusted using a Hemocytometer.

### 2.2. Animals and Experimental Design

Conventional C57BL/6 mice (7-week-old, male, SPF) were obtained from SLAC Jingda Laboratory Animal company, Changsha, China. Five mice were randomly allocated to each cage and housed in the animal facility under standard conditions (temperature: 22 ± 2 °C, humidity: 50 ± 15%, 12 h/12 h light–dark cycle).

After a two-week adaptation period, the animals were randomly divided into six groups based on their diet (*n* = 12/group): NC group (normal control), M group (model), HD group (dead-spore suspension of *E. amstelodami* H-1), HLH group (10^5^ spores/mL live-spore suspension of *E. amstelodami* H-1), HLL group (10^2^ spores/mL live-spore suspension of *E. amstelodami* H-1), and SLH group (10^5^ spores/mL live-spore suspension of *E. amstelodami* S-6). The experimental scheme was as follows: in the HD group, the spore suspension of *E. amstelodami* H-1 underwent sterilization at 121 °C for 20 min; mice in groups NC and M received 400 μL of the sterile water by oral gavage for 30 days, while 400 μL of the corresponding spore suspension was administered to the other groups—HD, HLH, HLL, and SLH—by oral gavage for 30 days. In the modeling stage (7 days), the NC group was given drinking water without DSS, whereas the other groups received 3% (*w*/*v*) DSS (36–50 kDa, MP Biomedicals, Santa Ana, CA, USA).

The mice’s body weight was monitored weekly throughout the protection period. Blood and fecal samples were collected before sacrificing the mice. On day 38, each mouse was sacrificed. Measurements were taken for colon length, and tissues from the colon, liver, and cecal contents were collected, rapidly frozen in liquid nitrogen, and stored at −80 °C for further analysis.

During the modeling stage, the body weight was assessed daily. All experimental procedures adhered to the guidelines outlined in the Guide for the Care and Use of Laboratory Animals: Eighth Edition, ISBN-10: 0-309-15396-4. The animal experimentation protocol was approval by the Animal Ethics Committee of Shangluo University. Surgeries on animal subjects were performed under anesthesia to minimize suffering.

### 2.3. Disease Activity Index (DAI) Assessment

During the modeling stage, the body weight, stool characteristics, and bloody feces were measured daily. Fecal occult blood test kits were utilized to examine stools from individual mice (C027-1-1, Nanjing Jiancheng Bioengineering Institute, Nanjing, China). Body weight loss was calculated as the percentage difference between the initial body weight on day 1 and the weight on each subsequent day. The Disease Activity Index (DAI) was determined by incorporating parameters such as body weight loss, diarrhea, and bleeding outcomes (Appendix A) [24].

### 2.4. Histopathological Analyses of the Livers of UC Mice

Mouse liver samples (about 1.5 cm^2^) were freshly isolated, fixed in 4% paraformaldehyde for 24 h, embedded in paraffin, and sectioned for hematoxylin and eosin (H&E) staining and Oil-Red staining to assess pathological histology.

### 2.5. Ultra-Structural Analysis of Colonic Tissue

For ultrastructural analysis, fresh colon tissues were cut into small pieces and initially fixed in a 2.5% glutaraldehyde solution at 4 °C. Subsequently, the tissues underwent re-fixation using 1% osmium tetroxide solution, followed by dehydration using a series of ethanol solutions with varying concentrations. A segment of the processed colon samples was then affixed to stubs, gold-coated, and analyzed using a scanning electron microscope (SEM). The remaining sections of the colon tissues subjected to treatment were impregnated with epoxy resin to produce ultrathin sections. These sections were subsequently stained with uranium acetate and aluminum citrate, then examined by transmission electron microscopy (TEM) [25]. The microvilli, tight junctions (TJ), and the other ultrastructures in the colonic epithelium were examined by SEM and TEM with a computer-aided image analysis system.

### 2.6. Measurement of Inflammatory Biomarkers by ELISA

After storage at 4 °C overnight, the serum sample was separated by centrifugation at 3000 rpm for 15 min at 4 °C. The concentrations of IL-6, IL-1β, IL-10, and TNF-α were determined using ELISA kits (Nanjing Jiancheng Technology Co., Ltd., Nanjing, China) in accordance with the manufacturer’s instructions. All experiments were repeated with three biological replicates.

### 2.7. Quantitative Reverse Transcription-Polymerase Chain Reaction (qRT-PCR) Analyses

Colon tissue total RNA was extracted using TRIzol reagent (Invitrogen Corporation Life Technologies, Carlsbad, CA, USA). The normalized RNA was calculated using an Ultramicro ultraviolet visible spectrophotometer (Darmstadt, Germany) and underwent reverse transcription using a UEIris RT mix with DNase (All-in-One) (US Everbright Inc., Changzhou, China) according to the protocol of the manufacturer. The PCR primer sequences are shown in Appendix A. Quantitative real-time RT-PCR (qRT-PCR) was conducted using a Bio-Rad CFX96 RealTime PCR Detection System (Bio-Rad, Hercules, CA, USA) with 2×SYBR Green qPCR Master Mix (US Everbright Inc., Changzhou, China), following the established procedure [26]. The calculation of relative gene expression between GAPDH and various target genes was determined using the Cycle threshold (Ct) value [27].

### 2.8. Gut Microbiota Analysis

Mice feces were promptly collected and snap-frozen with liquid nitrogen. All instruments used in this procedure underwent sterilization at 121 °C for 20 min. All group samples were stored at −80 °C until needed. Microbial DNA was extracted from mice feces using the E.Z.N.A.^®^ soil DNA Kit (Omega Biotek, Norcross, GA, USA), following the manufacturer’s instructions. The purity and concentration of the extracted DNA were assessed using a NanoDrop 2000 UV–vis spectrophotometer (Thermo Scientific, Wilmington, NC, USA). The V3–V4 hypervariable region of bacteria was amplified using primer pairs 338F (5′-ACTCCTACGGGAGGCAGCAG-3′) and 806R (5′-GGACTACHVGGGTWTCTAAT-3′). The ITS1-ITS2 hypervariable region of fungi was amplified with primer pairs ITS1F (5′-CTTGGTCATTTAGAGGAAGTAA-3′) and ITS2R (5′-GCTGCGTTCTTCATCGATGC-3′). PCR amplification products were retrieved through 2% agarose gel electrophoresis and purified via an AxyPrep DNA Gel Extraction Kit (Axygen Biosciences, Union City, CA, USA). After quantification, the purified PCR products underwent paired-end read stitching and were sequenced on a Miseq PE300 platform (Illumina, San Diego, CA, USA) using the protocol provided by Majorbio Bio-Pharm Technology Co., Ltd. (Shanghai, China).

### 2.9. Non-Targeted Fecal Metabonomics

Approximately 50 mg of fecal sample was weighed accurately, and metabolites were extracted by 400 µL methanol:water (4:1, *v*/*v*) solution. The mixture was allowed to settle at −20 °C and processed with a high-throughput tissue crusher Wonbio-96c (Shanghai wanbo biotechnology Co., Ltd., Shanghai, China), operating at 50 Hz for 6 min, followed by vortex for 30 s and ultrasound at 40 kHz for 30 min at 5 °C. After placing the samples at −20 °C for 30 min, proteins were precipitated. Following centrifugation at 13,000× *g* at 4 °C for 15 min, the supernatant was meticulously transferred to sample vials for LC–MS/MS analysis.

Metabolite chromatographic separation was conducted on an ExionLCTMAD system (AB Sciex, Boston, MA, USA) featuring an ACQUITY UPLC HSS T3 column (100 mm × 2.1 mm i.d., 1.8 µm; Waters, Milford, CT, USA). The mobile phases included 0.1% formic acid in water with formic acid (0.1%) (solvent A) and 0.1% formic acid in acetonitrile:isopropanol (1:1, *v*/*v*) (solvent B). The solvent gradient varied as follows: from 0 to 3 min, 95% (A): 5% (B) to 80% (A): 20% (B); from 3 to 9 min, 80% (A): 20% (B) to 5% (A): 95% (B); from 9 to 13 min, 5% (A): 95% (B) to 5% (A): 95% (B); from 13 to 13.1 min, 5% (A): 95% (B) to 95% (A): 5% (B); from 13.1 to 16 min, 95% (A): 5% (B) to 95% (A): 5% (B) for system equilibration. The sample injection volume was 20 uL, and the flow rate was set at 0.4 mL/min. The column temperature was maintained at 40 °C. During analysis, all samples were stored at 4 °C. The UPLC system was coupled with a quadrupole-time-of-flight mass spectrometer (Triple TOFTM5600+, AB Sciex, Boston, MA, USA) equipped with an electrospray ionization (ESI) source operating in positive mode and negative mode. Optimal conditions were set as follows: source temperature, 500 °C; curtain gas (CUR), 30 psi; both Ion Source GS1 and GS2, 50 psi; ion-spray voltage floating (ISVF), −4000 V in negative mode and 5000 V in positive mode; declustering potential, 80 V; a collision energy (CE), 20–60 V rolling for MS/MS. Data acquisition utilized the Data Dependent Acquisition (DDA) mode, covering a mass range of 50–1000 *m*/*z*.

Following UPLC–TOF/MS analyses, raw data were imported into Progenesis QI 2.3 (Nonlinear Dynamics, Waters, Boston, MA, USA) for peak detection and alignment. The preprocessing results yielded a data matrix containing retention time (RT), mass-to-charge ratio (*m*/*z*) values, and peak intensity. Metabolic features detected in at least 80% of any sample set were preserved. After filtering, metabolite values below the lower limit of quantitation were imputed with minimum values, and each metabolic feature was normalized by the sum. Data quality control (QC) employed the internal standard for reproducibility, with metabolic features exhibiting a relative standard deviation (RSD) of QC > 30% being excluded. Post-normalization and -imputation, log-transformed data underwent statistical analysis to discern significant differences in metabolite levels between comparable groups. Mass spectra of these metabolic features were matched using the accurate mass, MS/MS fragments spectra, and isotope ratio differences, searching in reputable biochemical databases such as the Human Metabolome database (HMDB) (http://www.hmdb.ca/) and Metlin database (https://metlin.scripps.edu/) accessed on 19 January 2022. Specifically, the mass tolerance between the measured *m*/*z* values and the exact mass of the components of interest was ±10 ppm. Metabolites with MS/MS confirmation were considered confidently identified only if the MS/MS fragment score exceeded 30; otherwise, metabolites received tentative assignments.

### 2.10. Statistical Analysis

Principal component analysis (PCA) was employed to identify general distribution trends in the samples. Distinctions between various groups were emphasized through orthogonal partial least squares discriminant analysis (OPLS-DA). The selection of differentially expressed metabolites (DEMs) relied on a VIP value exceeding 1 for the first principal component of the OPLS-DA model and a *p*-value below 0.05 for the *t* test [28].

## 3. Results

### 3.1. E. amstelodami Ameliorated DSS-Induced Colitis Mice

DSS-induced UC was induced in mice to assess the role of *E. amstelodami* in the pathogenesis of colitis (Figure 1). The results indicated that *E. amstelodami* supplementation (including HD, HLH, and HLL groups) significantly (*p* < 0.01) increased colon length (Figure 1A,B). During the protection period (Figure 1C), the weight of mice in the HD, HLH, HLL, and SLH groups showed an upward trend, consistent with the NC group, suggesting that *E. amstelodami* may be non-toxic, regardless of being live or dead. Between the 19th and 25th days, the weight of mice in the HLL group was notably (*p* < 0.05) increased compared with the NC group, which suggested that HLL treatment might have a superior protective effect against colitis. The DAI score showed no significant difference between the M, HLH, HLL, and SLH groups (Figure 1E), whereas HD treatment significantly lowered the DAI score compared to the M group. Moreover, compared with the NC group, the liver index in the HLL and SLH groups was significantly reduced (*p* < 0.05), aligning with previous literature reports [29].

### 3.2. HLL and SLH Administration Improved the Ultrastructure of Colonic Epithelium in Mice

The morphological damages were evident in the colon tissues of mice across various groups. Compared to the NC group, the colon in the M group exhibited atrophy and congestion ulcers. In contrast, the other treatment groups (HD, HLH, HLL, and SLH) significantly reduced the intestinal damage and increased the length of colon tissues based on their visual characteristics (Figure 1A). To further explore the protective effect of *E. amstelodami* intervention against gut injury, we selected fresh colon tissues, which were examined for the ultrastructure of the colonic epithelium (Figure 1G,H). The microvilli in the colon of the NC group were tightly and neatly arranged without any damage or shortening. The tight junctions (TJs) were prominently visible among the intestinal epithelial cells. DSS treatment notably disrupted the microvilli structure on the colon surface, leading to disorganized, loosely arranged, and collapsed microvilli, adherence to abnormal entities, and shortened TJs (Figure 2a,b). The administration of HLL and SLH notably increased the length of TJs, reduced microvilli damage, and expanded the intercellular space. These findings suggest that the supplementation of HLL and SLH could improve colon injury induced by DSS. Consistent with these results, it was observed that HLL and SLH treatment lowered the content of pro-inflammatory cytokines (IL-1β, IL-6, and TNF-α) in serum compared with the M group (Figure 1I–L). Notably, HLL significantly (*p* < 0.05) decreased the levels of IL-1β, IL-6, and TNF-α compared to the M group, while anti-inflammatory cytokine IL-10 was significantly (*p* < 0.05) increased by HLH supplementation. These data also confirmed that DSS-induced colitis was alleviated by the intervention of HLL and SLH.

### 3.3. HLL and SLH Supplementation Alleviates Liver Injury by DSS in Mice

As indicated in Figure 1F, compared to the NC group, the liver index of DSS-induced mice increased, whereas HLL and SLH treatment significantly suppressed the DSS-induced gain in liver index (*p* < 0.05). Additionally, relative to the NC group, the M group exhibited increased hepatic lipid accumulation characterized by numerous small lipid droplets, fatty vacuolization, and inflammatory cell infiltration, suggesting that DSS contributes to increased fat accumulation and inflammatory cell proliferation (Figure 2a,b). Hepatocyte lipid accumulation leads to liver metabolic dysfunction and steatosis in the liver. In the livers of the HLL and SLH groups, fewer fat droplets were observed compared to the M group, more closely resembling the NC group with fewer inflammatory cells. This indicates that HLL and SLH intervention effectively suppressed DSS-induced hepatic fat accumulation. The SLH group showed minimal steatosis and inflammatory foci, displaying only mild lipid droplet accumulation compared to the M group (Figure 2b). These results imply that SLH reduces the impact of DSS on hepatocytes, curbing inflammatory infiltration and aiding the recovery of hepatocyte structure. Some pro-inflammatory cytokines in DSS-induced mice can be transported to the liver via the gut–liver axis [30]. Thus, we further measured the expression level of pro-inflammatory cytokines in the liver was further measured (Figure 2L–O). The gene expression of pro-inflammatory cytokines IL-1β and TNF-α was significantly (*p* < 0.05) downregulated by HLL and HLH supplementation, while anti-inflammatory cytokine IL-10 was significantly upregulated, suggesting that HLL and HLH treatments can effectively alleviate liver inflammation to some extent. Notably, HLL intervention proved more effective, as evidenced by the more pronounced downregulation of IL-6 gene expression compared to HLH.

### 3.4. HLL Treatment Significantly Down-Regulated the Expression of Inflammatory Signal Pathway Related-Gene in Mice Liver

In pathological conditions, TRLs play a role in infectious colitis [31]. Thus, mechanisms are established to regulate the activation of TLR signals. The activation of TLR4 by components of bacterial cell walls initiates the activation of the downstream target genes that encode inflammation-related proteins, including NF-κBp65, COX-2, Nrf2, et al. Following the observations, we focused on HLL and HLH treatment groups for further investigation. As shown in Figure 2A–K, compared to the M group, the mRNA expression levels of inflammatory molecules in the liver, including TLR4, p65, Nrf2, COX-2, and iNOS, were significantly (*p* < 0.05) downregulated following HLL intervention, suggesting that HLL treatment effectively alleviated DSS-induced inflammation in the liver. Additionally, the NLRP3 inflammasome could be a target in the progression of UC [32]. Exploring anti-inflammatory mechanisms of *E. amstelodami*, we found that HLL supplementation significantly downregulated the gene expression of NLRP3 (*p* < 0.05), ASC (*p* < 0.05), and caspase-1 (*p* < 0.05) compared to the M group, indicating a notable improvement in liver inflammation.

Moreover, DSS-induced colitis was reported to suppress the gene expression of bile acid receptors (farnesoid-X-receptor, FXR; pregnane-X-receptor, PXR; and G-protein-coupled-receptor, TGR5) through the gut–liver axis, leading to liver injury [33]. Reduction of intestinal inflammation leads to liver improvement, as evidenced by the restoration of bile acid receptor levels. In our study, the gene expression of three key bile acid receptors (FXR, PXR, and TGR5) in the M group was lower than in the NC group, implying that intestinal inflammation adversely affected liver health, consistent with the literature [6]. Surprisingly, the mRNA expression levels of FXR and TGR5 were significantly (*p* < 0.05) upregulated by HLL intervention. Although the difference in PXR gene expression between the M and HLL groups was not significant, the mean value of PXR gene expression in the HLL group (0.985 ± 0.157) was notably higher than that in the M group (0.796 ± 0.025). These findings suggest that HLL treatment improved liver health by modulating the expression of bile receptors in the gut–liver axis.

### 3.5. HLH and HLL Treatment Modulates Bacterial Community Structure in DSS-Induced Mice

To further explore how *E. amstelodami* treatment affects colitis through the modulation of gut microbiota composition, we assessed the bacterial community structure changes in mice treated with *E. amstelodami* and in control groups via 16s rRNA gene amplicon sequencing. In the four groups of samples (NC, M, HLH, and HLL, *n* = 6), a total of 976,314 effective bacterial sequences were identified, spanning 414,859,629 bp. The rarefaction curves (Appendix A) plateaued, indicating that the sample sequencing depth was sufficient to capture the species diversity and saturate subsequent analysis. The rank-abundance curves (Appendix A) showed broad coverage on the horizontal axis, indicating reasonable richness and uniformity across the sample groups. Results from the Shannon index suggested that DSS (M group) induced a lower microbiota community diversity, while HLL and HLH treatment effectively restored the microbiota community diversity; nevertheless, no significant effect was observed on community richness (Appendix A). A total of 113 common bacterial species were identified in the four groups, and unique species in each group (NC, 168 species; M, 72 species; HLH, 100 species; and HLL, 95 species) were also identified. Among that, the bacterial composition of the HLL groups (152 species) was more diverse than that in the M (105 species) (Appendix A). NMDS analysis was conducted on bacterial communities across all samples, resulting in the dispersion of four sample groups into distinct quadrants, signifying notable variations in bacterial composition among the groups; the sample confidence ellipse of HLL and NC group were the closest, indicating that HLL treatment was more effective (Appendix A).

The bacteria belonged to 7 phyla, 8 classes, 15 orders, 19 families, 20 genera, 23 species, and 32 operational taxonomic units (OTUs). The results of phylum distributions are shown in Figure 3(A1); *Proteobacteria*, *Bacteroidetes*, and *Firmicutes* were the primary bacteria among the four groups. The abundance of *Proteobacteria* in the M group (69.76%) was significantly higher than that in the NC group (27.65%); the abundance of *Bacteroidetes* (10.35%) and *Firmicutes* (17.58%) was significantly lower than that in the NC group (32.37%, 23.8%). There was a slight recovery after *E. amstelodami* supplementation. In both the HLH and HLL groups, the proportion of *Firmicutes/Bacteroidetes* was closer to the level of the NC group, and the abundance of *Proteobacteria* (39.52%, 30.41%) significantly decreased. Interestingly, the abundance of *Actinobacteria* increased after HLL intervention. At the class level (Figure 3(A2)), after DSS treatment, the abundance of γ-*Proteobacteria* increased, and the abundance of *Bacteroidia*, *Bacilli*, and *Verrucomicrobiae* significantly decreased compared with NC group. However, after supplementation with *E. amstelodami* (HLH and HLL), the opposite trend was shown, that is, the abundance of γ-*Proteobacteria* decreased, while the abundance of *Bacteroidia*, *Bacilli*, and *Verrucomicrobiae* significantly increased, to values closer to the level in the NC group. Notably, the abundance of *Actinobacteria* increased significantly after HLL treatment. The results of the genus distributions are shown in Figure 3(A3). Compared with the NC group, in the M group, the abundance of *Escherichia-Shigella* and *Enterococcus* significantly increased, the abundance of norank_f_*Muribaculaceae*, *Lactobacillus*, *Bifidobacterium*, *Akkermansia*, unclassified_f_*Lachnospiraceae*, *Candidatus_Saccharimonas*, norank_f_o_*Clostridia*_UCG-014, and *Lachnospiraceae*_NK4A136_group markedly decreased. Nevertheless, *E. amstelodami* treatment (HLH and HLL) significantly enriched gut bacterial community composition and diversity. It is worth noting that, after supplementation with HLL, the abundance of beneficial bacteria *Lactobacillus*, *Bifidobacterium*, *Akkermansia*, and *Faecalibaculum* in the colitis mice increased greatly, the abundance of harmful bacteria *Escherichia-Shigella* reduced evidently. Thus, the structure of intestinal bacterial flora after HLL treatment was closer to that in the NC group, suggesting that HLL supplementation was effective to treat DSS-induced gut microbiota imbalance in mice.

Differences in the top 15 bacterial genera among different groups were identified by STAMP analysis (*Wilcoxon*-test). Compared with the NC group (Figure 3(B1)), the abundance of *Escherichia-Shigella*, *Bacteroides*, and *Enterococcus* in M group was significantly (*p* < 0.05) increased, and the abundance of beneficial bacteria *Lactobacillus* and *Akkermansia* was evidently (*p* < 0.01) reduced, suggesting that DSS induction substantially altered gut bacterial structure. Furthermore, HLL supplementation significantly increased the abundance of beneficial bacterial *Bifidobacterium* and *Akkermansia* (Figure 3(B2)), and HLH intervention also markedly increased the abundance of beneficial bacterial *Akkermansia* (Figure 3(B3)), indicating that HLL and HLH treatment can restore the imbalance of gut bacterial community structure caused by DSS to a certain extent and that HLL intervention was more effective. To assess the particular bacterial taxa linked to DSS or *E. amstelodami* intervention, we conducted additional LEfSe analysis to pinpoint statistically significant biomarkers within the gut microbiota of different groups. The distribution histogram of LDA scores (>3) is presented in Figure 3C. The specific genera in the M group mainly included f_*Enterococcaceae* and g_*Enterococcus*, while the genera specific to the HLH group included g_*Bacteroides*, f_*Bacteroidaceae*, f_*Staphylococcaceae*, and o_*Staphylococcales*, and the genera specific to the HLL group included o_*Erysipelotrichales* 4.72960284201, f_*Erysipelotrichaceae*, p_*Actinobacteriota*, f_*Bifidobacteriaceae*, o_*Bifidobacteriales*, g_*Bifidobacterium*, c_*Actinobacteria*, g_*Faecalibaculum*, o_*Clostridiales*, f_*Clostridiaceae*, and g_*Clostridium*_sensu_stricto_1.

### 3.6. HLL Supplementation Alteres Fungal Community Structure in DSS-Induced Mice

The changes in the fungal community structure in mice treated with *E. amstelodami* and in control groups were also assessed, this time using ITS rRNA gene amplicon sequencing. Across the four groups (NC, M, HLH, and HLL, *n* = 6), 1,372,459 effective fungal sequences were identified, totaling 312,794,115 bp. The sobs index curves for all samples (Appendix A) confirmed that the volume of sequencing data was sufficient for thorough analysis, and the grading curves (Appendix A) indicated that the samples exhibited reasonable richness and evenness. Results from the Shannon index revealed that the fungal community diversity in the M group was significantly (*p* < 0.05) lower than in the NC group, whereas diversity was notably (*p* < 0.05) restored following HLL treatment. A Venn diagram (Appendix A) identified 64 OTUs that were common across the 4 groups. The fungal community composition unique to the M group (208 species) was significantly more abundant than that in the HLH (99 species) and HLL (180 species) groups, suggesting a prevalence of potentially harmful fungal species in the M group. Further, PCA analysis differentiated the four sample groups into distinct quadrants, with the NC and HLL groups clustering in the same quadrant, suggesting differences in fungal community composition between these groups.

The fungi belonged to 4 phyla, 9 classes, 13 orders, 14 families, 17 genera, 20 species, and 23 OTUs. At the phylum level, DSS exposure induced a marked increase in the abundance of *Basidiomycota* and a decrease in *Ascomycota* compared with the NC group. Among them, HLL administration significantly reversed this trend in *Basidiomycota* and *Ascomycota* (Figure 4(A1–A3)). At the order level (Figure 4(A2)), DSS exposure led to an increase in the abundance of *Trichosporonales* and *Saccharomycetales* and a decrease in *Eurotiales*, *Hypocreales*, *Wallemiales*, *Capnodiales*, unclassified_k_*fungi*, *Tremellales*, *Pleosporales*, *Sporidiobolales,* and *Mortierellales*. Compared with the M group, HLL supplementation significantly increased the abundance of *Eurotiales*, *Hypocreales*, *Wallemiales*, *Conioscyphales*, *Capnodiales*, *Sporidiobolales*, *Tremellales*, *Pleosporales*, and unclassified_k_*fungi* and decreased *Trichosporonales*, *Saccharomycetales*, and *Microascales*. It can be seen that the overall fungi community structure in the HLL group was closer to the NC group. At the genus level (Figure 4(A3)), in the HLL group, the fungal community distribution was more abundant and diverse, which was more consistent with that in the NC group, indicating that HLL administration is effective to restore gut fungi flora disturbance caused by DSS. Specifically, compared with the M group, except for *Cutaneotrichosporon* and *Saccharomyces*, the abundance of other fungal genera, including *Aspergillus*, unclassified_f_*Aspergillaceae*, *Penicillium*, *Talaromyces*, *Wallemia*, *Fusarium*, *Trichoderma*, *Cladosporium*, *Eurotium*, unclassified_k_*Fungi*, *Candida*, unclassified_o_*Conioscyphales*, *Mortierella*, and *Neocosmospora*, increased in the HLL group.

Moreover, based on STAMP analysis, it can be clearly seen that DSS exposure significantly (*p* < 0.01) increased the abundance of *Cutaneotrichosporon* and *Saccharomyces* compared with the NC group (Figure 4(B1)). Compared with the M group, HLH intervention decreased the abundance of *Cutaneotrichosporon* and increased the abundance of unclassified_f_*Aspergillaceae* and *Wallemia* with no significant difference (Figure 4(B2)). By contrast, HLL treatment significantly (*p* < 0.01) reduced the abundance of *Cutaneotrichosporon* and *Saccharomyces*, and markedly (*p* < 0.05 or *p* < 0.01) increased the abundance of *Aspergillus*, unclassified_f_*Aspergillaceae*, *Trichoderma*, *Wallemia*, *Eurotium*, *Cladosporium*, unclassified_k_*Fungi*, and unclassified_o_*Conioscyphales* (Figure 4(B3)). Thus, HLL supplementation is more effective in ameliorating DSS-induced gut fungal community disturbance.

To further identify the characteristic species biomarkers in each group, LDA analysis was performed to display the effect of differential species on differences between groups. As shown in Figure 4C, g_unclassified_f_*Aspergillaceae* can be considered as a characteristic species of the HLH group. Similarly, the biomarker species in the HLL group can be considered as p_*Ascomycota*, g_*Aspergillus*, c_*Sordariomycetes*, o_*Hypocreales*, f_*Hypocreaceae*, g_*Trichoderma*, g_*Penicillium*, f_*Trichocomaceae*, and g_*Talaromyces*; the biomarker species in the M group can be considered as g_*Cutaneotrichosporon*, o_*Trichosporonales*, f_*Trichosporonaceae*, c_*Tremellomycetes*, p_*Basidiomycota*, g_unclassified_f_*Microascaceae*, f_*Psathyrellaceae*, and g_*Scopulariopsis*.

### 3.7. Correlations between Inflammation-Related Gene Expression and Gut Microorganisms

To examine the connections between the modified microbial community and colitis-related parameters, Spearman’s rank correlation analysis was conducted. Our findings indicated that microbial community were correlated with inflammatory cytokine and specific inflammatory pathway-related gene expression (Figure 5). For bacterial communities, pro-inflammatory cytokines IL-6, IL-1β, and TNF-α, were positively related to the harmful bacterial *Enterococcus*, *Staphylococcus*, and *Escherichia-Shigella*, which were enriched in the M group, and were negatively correlated with the beneficial bacterial *Bifidobacterium* and *Akkermansia*, which were enriched in the HLL and NC group. Anti-inflammatory cytokine IL-10 was negatively related to *Enterococcus* and *Staphylococcus* and positively correlated with *Bacteroides*, *Dubosiella*, and *Parasutterella*. Moreover, some inflammatory signaling pathway-related genes, including NLRP3 inflammasome (ASC, Caspase-1, and NLRP3) and NF-κB family (COX-2, iNOS, Nrf2, p65, and TLR4), in the liver were significantly negatively correlated with *Bifidobacterium*, *Akkermansia*, and *Enterorhabdus* and were markedly positively related to *Escherichia-Shigella*, *Enterococcus*, and *Staphylococcus*. Consistent with the above trend, bile acid receptors (FXR, PXR, and TGR5) were positively correlated to the beneficial bacterial *Bifidobacterium*, *Akkermansia*, and *Faecalibaculum* and negatively related to *Enterococcus* and *Staphylococcus*. These findings suggested that the increase in harmful bacteria by DSS-induced leads to the production of pro-inflammatory cytokines and the expression of related inflammatory genes, thereby aggravating the degree of inflammation in the colon or liver. For fungal communities, pro-inflammatory cytokines (IL-6, IL-1β, and TNF-α) and inflammation-related genes (ASC, Caspase-1, NLRP3, COX-2, iNOS, Nrf2, p65, and TLR4) were both negatively correlated with *Aspergillus*, unclassified_f_*Aspergillaceae*, *Penicillium*, *Talaromyces*, *Wallemia*, *Fusarium*, *Trichoderma*, *Cladosporium*, *Eurotium*, unclassified_k_*Fungi*, *Candida*, and *Cryptococcus*_f_*Tremellaceae*, which were enriched in the HLL group, and were positively related to *Cutaneotrichosporon*, *Saccharomyces*, and *Exophiala*, which were enriched in the M group. Nevertheless, both anti-inflammatory IL-10 and bile acid receptors (FXR, PXR, and TGR5) were obviously positively related to *Aspergillus*, *Talaromyces*, *Wallemia*, *Trichoderma*, *Cladosporium*, *Eurotium*, and unclassified_o_*Conioscyphales* and were negatively correlated to *Cutaneotrichosporon*. These findings indicate that HLL treatment reduced DSS-induced systemic inflammation in mice by increasing the diversity and abundance of the fungal community such as *Aspergillus*, *Penicillium*, *Talaromyces*, unclassified_f_*Aspergillaceae*, *Trichoderma*, *Wallemia*, *Eurotium*, *Fusarium*, and *Cladosporium*.

### 3.8. HLL Intervention Regulates Gut Metabolism in Colitis Mice

Principal component analysis (PCA) was executed using the peaks extracted from all experimental samples and QC samples (Appendix A). QC samples exhibited close clustering in both positive and negative ion modes, indicating robust repeatability in the experiment. Based on the PCA analysis of all samples, we found that four groups all clustered into different quadrants, indicating significant differences among groups. Then, we focused on the differences between DSS-induced colitis mice and those exposed to the HLL administration diet. The OPLS-DA of the samples from the M group and HLL group showed a clear distinction between two groups (Appendix A), and the positive and negative ion modes were both sorted and validated (R2 = (0, 0.6481), Q2 = (0, −0.3613); R2 = (0, 0.5318), Q2 = (0, −0.6018)), indicating that the model fit relatively well. In total, 251 metabolites in the HLL&M group were identified based on the HMDB database, among which 94 and 157 metabolites were detected in positive and negative ion modes, respectively, including 68 organic acids and derivatives, 66 lipids and lipid-like molecules, 33 organoheterocyclic compounds, 28 phenylpropanoids and polyketides, 24 organic oxygen compounds, 16 benzenoids, 9 nucleosides, nucleotides, and analogues, 3 organic nitrogen compounds, 2 alkaloids and derivatives, 1 organic 1,3-dipolar compound, and 1 organosulfur compound (Figure 6A).

Based on the KEGG functional pathway, all metabolites in the HLL&M group were mainly classified into three KEGG pathway levels, including metabolism, organismal systems, and human diseases, among which a total of 115 metabolites were involved in amino acid metabolism, biosynthesis of other secondary metabolites, carbohydrate metabolism, chemical structure transformation maps, energy metabolism, lipid metabolism, metabolism of cofactors and vitamins, metabolism of other amino acids, metabolism of terpenoids and polyketides, nucleotide metabolism, and xenobiotics biodegradation and metabolism (Figure 7A). Furthermore, VIP (Variable Importance for the Projection) > 1 and *p* < 0.05 were used to screen out differential metabolites between the HLL and M groups. Figure 6B shows all differential metabolites in the two groups. Among which, the six most differentially expressed metabolites were identified as 1—Alanyl-Aspartic acid; 2—2,5,8-trihydroxy-6-methoxy-2-methyl-3H-benzo[g]chromen-4-one; 3—(4-ethyl-2-hydroxy-6-methoxyphenyl)oxidanesulfonic acid); 4—Pyren-1-ol; 5—N6-Carbamoyl-L-threonyladenosine; and 6—6-(carboxymethoxy)-3,4,5-trihydroxyoxane-2-carboxylic acid. Moreover, a heatmap was used to visualize the hierarchical clustering of the top 30 most significantly altered metabolites and metabolites related to the important pathways (Figure 6C). After treatment with HLL, the expression of metabolites Alanyl-Aspartic acid, (2Z,4′Z)-2-(5-Methylthio-4-penten-2-ynylidene)-1,6-dioxaspiro [4.4]non-3-ene, {4-[3-oxo-3-(2,4,6-trihydroxyphenyl)prop-1-en-1-yl]phenyl}oxidanesulfonic acid, and (4-ethyl-2-hydroxy-6-methoxyphenyl)oxidanesulfonic acid, 2-[(4,6-diamino-1,3,5-triazin-2-yl)sulfanyl]ethanesulfonic acid were significantly (*p* < 0.001) down-regulated, and 5,7-dihydroxychromen-2-one was obviously up-regulated compared with the M group. In addition, the KEGG topology analysis revealed that there were five primary disturbed pathways (*p* < 0.05) of feces, involving (1) D-Arginine and D-ornithine, (2) Alanine, aspartate, and glutamate metabolism, (3) Cutin, suberine, and wax biosynthesis, (4) Isoflavonoid biosynthesis, and (5) Histidine metabolism. Among these, after HLL treatment, metabolites involved in the five important metabolic pathways above, including D-ornithine, L-aspartic acid, and L-glutamate, were significantly increased; however, L-Arginine, 16-hydroxypalmitic acid, daidzein, glycitein, liquiritigenin, genistein, histidinal, and L-histidine were significantly reduced (Figure 7B).

### 3.9. Correlations between Differential Metabolites and Predominant Microorganisms

Changes in the gut microbiome affect the perturbation of its metabolites, which in turn affects the development of colitis. A correlation heatmap was utilized to depict the covariation between predominant genera of gut microbes and the modified differential fecal metabolites in the HLL&M group (Figure 7C,D). All metabolites were grouped into two main clusters according to the magnitude of the correlation between the metabolites. For bacterial communities, the abundance of g_*Bifidobacterium*, g_*Akkermansia*, g_*Dubosiella*, g_*Parasutterella*, g_unclassified_f_*Oscillospiraceae*, and g_norank_f_*Ruminococcaceae* was significantly positively related to the production of differential metabolites 1-(11Z-eicosenoyl)-glycero-3-phosphate, L-glutamate, acetylcholine, N-oleyl-leucine, phosphatidylethanolamine lyso alkenyl 18:1, [3-[2-aminoethoxy(hydroxy)phosphoryl]oxy-2-hydroxypropyl] (Z)-octadec-9-enoate, Creatine, LysoPC(22:6(4Z,7Z,10Z,13Z,16Z,19Z)), and PE(17:1(9Z)/0:0), and was obviously negatively correlated to the production of 2-[(2-amino-3-methylpentanoyl)amino]-3-methylpentanoic acid, N-acetyl-L-histidine, medicagenic acid, N2-(D-1-Carboxyethyl)-L-lysine, (R)-b-amino-isobutyric acid, L-leucyl-L-alanine, glycerophospho-N-oleoyl ethanolamine, and threoninyl-isoleucine. Meanwhile, for fungal communities, g_*Cutaneotrichosporon*, g_*Exophiala*, and g_*Saccharomyces* were positively correlated with the metabolites in cluster I and negatively correlated with the metabolites in cluster II; however, the other fungi (including g_*Aspergillus*, g_*Cryptococcus*_f_*Tremellaceae*, g_unclassified_f_*Dipodascaceae*, g_*Trichomonascus*, g_unclassified_f_*Aspergillaceae*, g_*Trichoderma*, g_*Wallemia*, g_*Eurotium*, g_*Cladodporium*, and g_unclassified_f_*Conioscyphales*) were positively associated with the metabolites in cluster II and negatively related to cluster I.

## 4. Discussion and Conclusions

Although the precise mechanism of ulcerative colitis is still unclear, numerous studies suggest that it is strongly related to the occurrence of inflammation, disturbance of the intestinal flora, and the resulting changes in the metabolic profile [13,17]. Thus, the aim was mainly to explore the protective mechanism of *E. amstelodami* against colitis from three aspects.

In our study, UC mice symptoms such as weight loss, colon shortening, and disrupted microvilli structure caused by DSS significantly improved after HLL treatment, which is currently closely related to gut microbiota imbalances such as an increase in beneficial microbes and a decrease in harmful microbes [10]. As our findings show, after DSS treatment, the gut microbiota structure was severely damaged compared to the NC group; however, after HLL treatment, the gut microbiota structure was evidently restored, showing a more similar microbiota structure, including both bacterial and fungal communities, to the NC group (Figure 3 and Figure 4). A decrease in *Firmicutes* and an increase in *Proteobacteria* were observed to be caused by DSS, consistent with the literature [34]. *Gammaproteobacteria* are Gram-negative bacteria that include a variety of pathogenic bacteria such as *Salmonella* and *Helicobacter*; their abnormal expansion is a potential signal of microbial dysbiosis and disease [35]. Consistent with the above point of view, our study found that DSS treatment markedly elevated the abundance of the harmful order *Proteobacteria*, but abundance was subsequently reversed by HLL treatment; meanwhile, it increased the abundance of beneficial orders *Bacteroidia*, *Bacilli*, *Clostridia*, and *Verrucomicrobiae* (Figure 3). Notably, *Verrucomicrobiae* contributed to upregulated expression of MUC2, protecting the gut barrier [36], which was increased by HLL supplementation. Similarly, HLL intervention significantly increased the abundance of beneficial bacterial genera *Bacteroides*, *Lactobacillus*, *Bifidobacterium*, *Akkermansia*, *Faecalibaculum*, and *Clostridium*_sensu_stricto_1 and reduced the harmful bacterial genera *Escherichia-Shigella*, *Enterococcus*, and *Staphylococcus*. Among these, *Lactobacillus*, *Bifidobacterium*, and *Clostridium*_sensu_stricto_1 have been considered as beneficial bacteria that improve gastrointestinal health by protecting the immune system. As Chen et al. have shown, the enrichment of *Lactobacillus*, *Bifidobacterium*, and *Clostridiales*_unclassified can ameliorate intestinal mucositis [37]. *Bacteroides* and *Akkermansia*, the two enriched genera, are known to play crucial roles in colitis or colitis-related diseases. *Bacteroides ovatus* monotherapy has shown consistent and effective outcomes in ameliorating colitis compared to traditional fecal microbiota transplantation [38]. Another type of *Bacteroides*, *Bacteroides vulgatus*, has been demonstrated to provide defense against colitis induced by Escherichia coli in IL-2-/- mice. Treatment with *Akkermansia muciniphila* has been shown to attenuate inflammation and protect the intestinal barrier [39]. Li et al. conjectured that Faecalibaculum is a beneficial gut bacteria that maintains the homeostasis of gut flora. The previous literature reported that *Escherichia-Shigella*, *Staphylococcus*, and *Enterococcus* were greatly increased in IBD animals, resulting in exacerbated intestinal inflammation [40,41]. And Zhu et al. found that the depletion of *Escherichia-Shigella* can alleviate colitis symptoms [42].

On the other hand, a lower percentage of *Ascomycota* and an increase in *Basidiomycota* at phylum level were also observed to be induced by DSS. At the order level, *Eurotiomycetes* and *Sordariomycetes* were reduced and *Tremellomycetes* was increased by DSS treatment; however, the above trend was reversed after HLL intervention. Moreover, at the genus level, a decrease in the abundance of *Cutaneotrichosporon* and *Saccharomyces* and an increase in abundance of *Aspergillus*, *Penicillium*, *Talaromyces*, *Wallemia*, *Trichoderma*, *Cladosporium*, and *Eurotium* in HLL group were observed. Among these, *Saccharomyces* is a beneficial strain that is widely used in food fermentation such as fermented wine; it can produce pleasant aroma and flavor [43]. The decrease in *Saccharomyces* can be interpreted as a result of nutrient competition with other fungi [44]. As is known, *Aspergillus*, *Penicillium*, *Talaromyces*, *Wallemia*, *Trichoderma*, *Cladosporium*, and *Eurotium* are the dominant genera in the fermentation process of Fuzhuan brick tea [45], which can secrete a variety of enzymes such as cellulase, glucoamylase, polyphenol oxidase, and pectinase and can thereby facilitate the degradation and transformation of macromolecular substances, including (but not limited to) cellulose, polysaccharide, protein, polyphenol, and amino acid [20,46,47]. Therefore, they can promote the breakdown of gut metabolites.

In recent times, the concept of the gut–liver axis has garnered significant attention due to inflammatory translocation and functional crosstalk between the intestine and the liver. DSS exposure can disrupt the intestinal barrier, increase intestinal permeability, and lead to the translocation of lipopolysaccharide (LPS), which aggravate the development of liver inflammation, manifested by overexpression of pro-inflammatory cytokines and oxidative stress. In our study, down-regulated pro-inflammatory cytokines (IL-6, IL-1β, TNF-α) and up-regulated anti-inflammatory cytokine IL-10 were observed in the HLL group (Figure 2). Furthermore, NLRP3 inflammasome family (ASC, Caspase-1, NLRP3) and TLR4-activated NF-κB pathway-related inflammatory response (TLR4, p65, Nrf2, and COX-2) was significantly restrained by HLL supplementation (Figure 5). As the literature has reported, NF-κB controls a large number of IBD-related inflammatory genes to regulate the intestinal barrier permeability and promote the expression of pro-inflammatory cytokines [41]. Miao et al. [5] reported that negative regulation of the inflammasome NLRP3 pathway is an important molecular basis for improving colitis. This is consistent with our findings, as the gut–liver axis links the liver and intestine through bile acid metabolism. The alleviation of intestinal inflammation leads to liver improvement manifested by restoration of bile acid receptor (FXR, PXR, TGR5) level [33]. Consistently, we observed that the gene expression level of bile acid receptor FXR and TGR5 was significantly enhanced by HLL treatment (Figure 2).

Untargeted metabolomic studies have elucidated extensive perturbations of gut metabolites associated with gut dysbiosis in UC. A multi-omics approach opens new doors for us to further explore the underlying mechanisms of colitis pathogenesis. In recent years, combined analysis of the gut microbiome and metabolome have been widely used in various studies on the treatment of colitis [48]. A growing number of studies have demonstrated that gut microbiome fluctuation directly alters the changes in the metabolic profile in fecal samples. The beneficial bacterial g_*Bifidobacterium* and g_*Akkermansia* were positively correlated with metabolites 1-(11Z-eicosenoyl)-glycero-3-phosphate, L-glutamate, acetylcholine, N-oleyl-leucine, phosphatidylethanolamine lyso alkenyl 18:1, [3-[2-aminoethoxy(hydroxy)phosphoryl]oxy-2-hydroxypropyl] (Z)-octadec-9-enoate, and PE(17:1(9Z)/0:0). They are mainly involved in amino acid metabolism, which is necessary for nutrient metabolism and the maintenance of the mucosal integrity and barrier function [49]. Simultaneously, we observed that g_*Bifidobacterium* and g_*Akkermansia* were significantly negatively correlated with pro-inflammatory cytokines and inflammatory gene expression (Figure 5). This suggests that beneficial bacteria g_*Bifidobacterium*, g_*Akkermansia*, and their metabolites may reduce colitis by suppressing the expression of inflammatory genes. More to the point, the potential beneficial fungi *Aspergillus*, unclassified_f_*Aspergillaceae*, *Penicillium*, *Talaromyces*, *Wallemia*, *Fusarium*, *Trichoderma*, *Cladosporium*, *Eurotium*, unclassified_k_*Fungi*, and *Cryptococcus*_f_*Tremellaceae* were positively associated with metabolites in cluster II. Among these, lipid metabolism (1-nonadecanoyl-glycero-3-phosphate, LysoPE, LysoPC, 1-(11Z-eicosenoyl)-glycero-3-phosphate, PE, PG, Phosphatidylethanolamine lyso alkenyl 18:1) and organic acids’ (and their derivatives’) metabolism (N-Oleyl-leucine, 4-Methylene-L-glutamine, pantothenic acid, L-glutamate) were two important metabolic pathways. Wang et al. reported that *Patrinia villosa* exhibited an obvious therapeutic effect on UC by regulating lipid metabolism [50]. Some organic metabolites, such as the biosynthesis of pantothenic acid, were reported to have anti-inflammatory effects [51]. Intriguingly, these potential beneficial fungi were also shown to be negatively correlated with pro-inflammatory cytokines and inflammatory gene expression in this study.

Taken together, the correlation analysis of the gut microbiome, metabolome, and inflammatory parameters presented the overall correlativity during HLL treatment. The profound changes in gut taxa, metabolites, and inflammatory markers provided a detailed insight into the therapeutic mechanism of HLL in UC. Xie et al. [52] reported that the intracellular polysaccharides in *Aspergillus cristatus* from Fuzhuan brick tea can alleviate symptoms of DSS-induced colitis in mice by repairing the intestinal barrier. And Lu et al. [53] noted that the polysaccharides in *Eurotium cristatum* from Fuzhuan brick tea can ameliorate colitis in mice by modulating the gut microbiota. This is consistent with our research findings. Accordingly, we believe that the correlation analysis of the gut microbiome, metabolome, and inflammatory markers can be attributed to the anti-inflammatory effects of intracellular and extracellular polysaccharides in *E. amstelodami* H-1 (10^2^ spores/mL live-spore suspension), when combined with the literature. The restoration of gut microbial community structure, which is deeply implicated in the inflammatory parameters’ and metabolites’ profiles, might be essential elements in the therapeutic mechanism of *E. amstelodami* on colitis.

## Figures and Tables

**Figure 1 nutrients-16-01178-f001:**
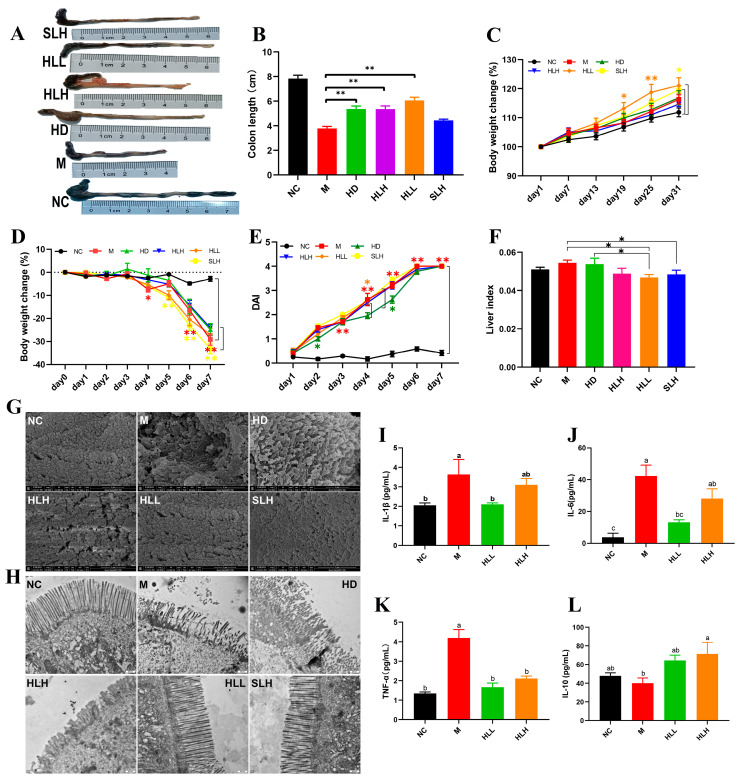
Effects of *E. amstelodami* on DSS-induced colitis mice. (**A**) Representative images of the mouse colon. (**B**) Colon length. (**C**) Body weight change during the protection period. (**D**) Body weight change during the modeling period. (**E**) DAI scores. (**F**) Liver index. The surface of the colonic epithelium under SEM (**G**) and TEM (**H**). The levels of (**I**) IL-1β, (**J**) IL-6, (**K**) TNF-α, and (**L**) IL-10 in the serum. Data are presented as the mean ± SEM. One asterisk (*) indicates *p* < 0.05, and two asterisks (**) indicate *p* < 0.01. Different lowercase letters (a, b, and c) indicate significant differences at the level of *p* < 0.05.

**Figure 2 nutrients-16-01178-f002:**
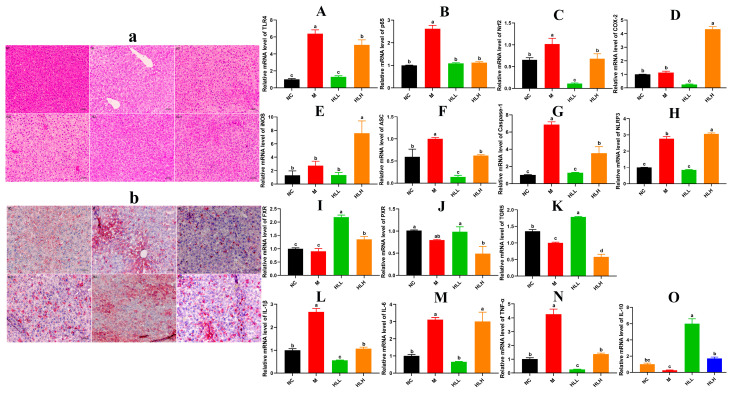
Effect of live-spore suspension of *E. amstelodami* H-1 (10^2^ spores/mL, HLL) intervention on liver health in mice. (**a**) H&E staining and (**b**) Oil-Red staining of the liver (200× magnification). The relative gene expression of key inflammatory cytokines (**A**) TLR4, (**B**) p65, (**C**) Nrf2, (**D**) COX-2, and (**E**) iNOS in the NF-κB signaling pathway. The relative gene expression levels of (**F**) ASC, (**G**) Caspase-1, and (**H**) NLRP3 in the NLRP3 signaling pathway. The relative gene expression levels of bile acid receptors in the gut–liver axis, including (**I**) FXR, (**J**) PXR, and (**K**) TGR5. The relative gene expression levels of (**L**) IL-1β, (**M**) IL-6, (**N**) TNF-α, and (**O**) IL-10 in liver. Data are presented as the mean ± SEM. Different lowercase letters (a, b, c, and d) indicate significant differences at the level of *p* < 0.05.

**Figure 3 nutrients-16-01178-f003:**
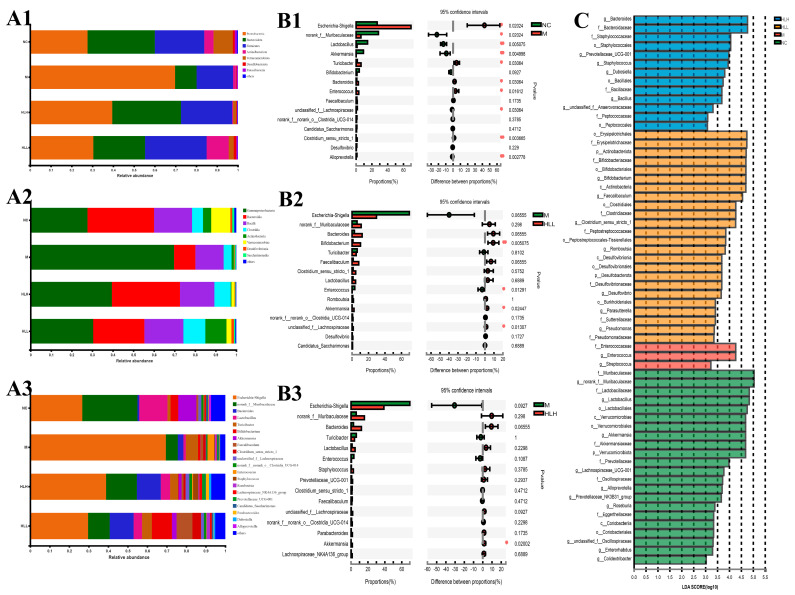
Bacterial community distribution among NC (normal control), M (model), HLL (10^2^ spores/mL *E. amstelodami* H−1), and HLH (10^5^ spores/mL *E. amstelodami* H−1) groups. The histogram of bacterial community distribution at the (**A1**) phylum, (**A2**) order, and (**A3**) genus levels in the four groups. Wilcoxon rank-sum test bar plot of bacterial communities at the genus level in (**B1**) NC vs. M, (**B2**) M vs. HLL, and (**B3**) M vs. HLH among four groups. One asterisk (*) indicates *p* < 0.05, and two asterisks (**) indicate *p* < 0.01. (**C**) Discriminative bacteria biomarkers with an LDA score > 3 between the four groups.

**Figure 4 nutrients-16-01178-f004:**
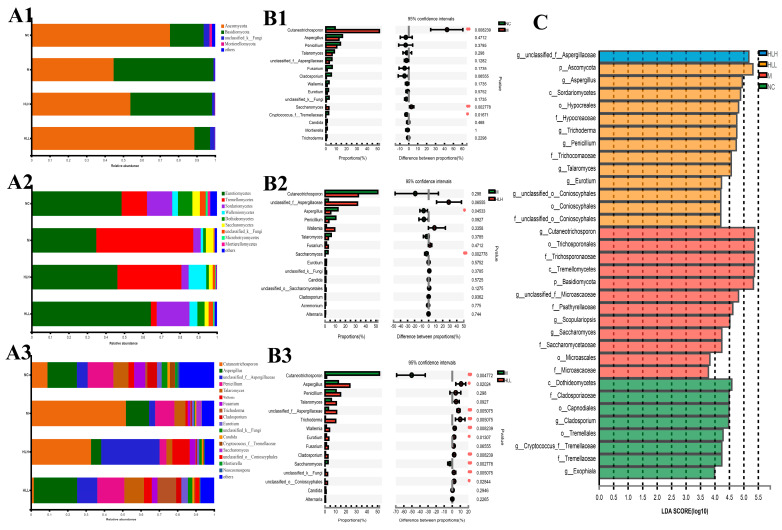
Fungal community distribution among NC (normal control), M (model), HLL (10^2^ spores/mL *E. amstelodami* H−1), and HLH (10^5^ spores/mL *E. amstelodami* H−1) groups. The histogram of fungal community distribution at the (**A1**) phylum, (**A2**) order, and (**A3**) genus levels in the four groups. Wilcoxon rank-sum test bar plot of fungal community at the genus level in (**B1**) NC vs. M, (**B2**) M vs. HLH, and (**B3**) M vs. HLL. One asterisk (*) indicates *p* < 0.05, and two asterisks (**) indicate *p* < 0.01. (**C**) Discriminative fungi biomarkers with an LDA score > 3.5 between the four groups.

**Figure 5 nutrients-16-01178-f005:**
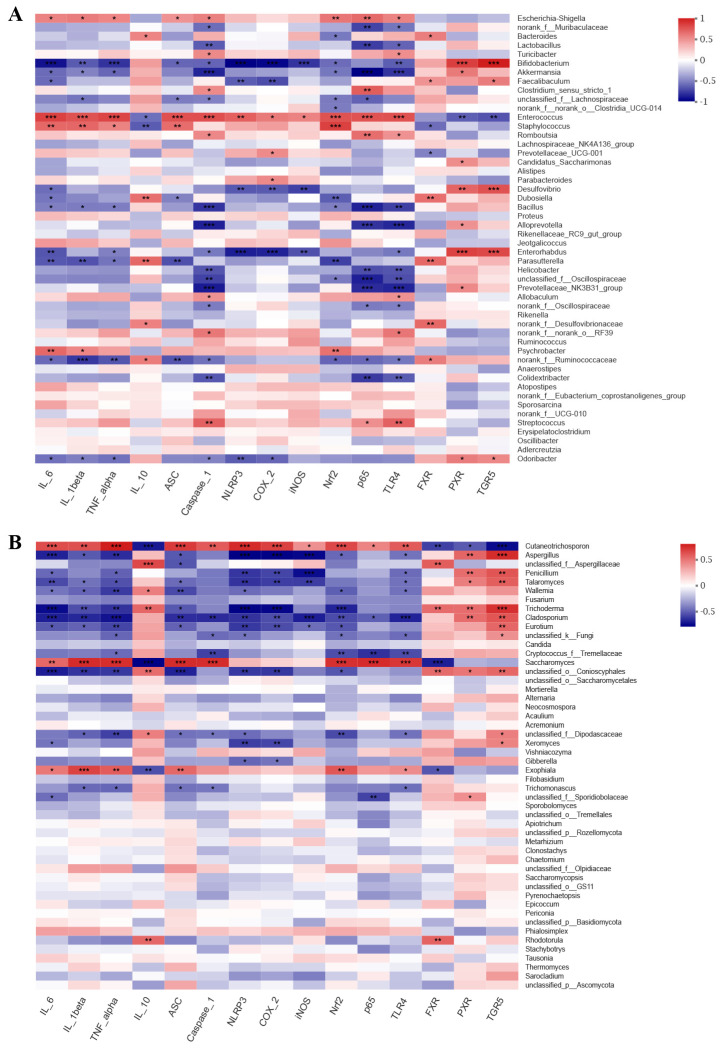
Correlation analysis between gut microbes ((**A**). bacteria, (**B**), fungi) and inflammatory−related markers. Blue indicates positive correlation, and red indicates negative correlation. One asterisk (*) indicates *p* < 0.05, two asterisks (**) indicate *p* < 0.01, and three asterisks (***) indicate *p* < 0.001.

**Figure 6 nutrients-16-01178-f006:**
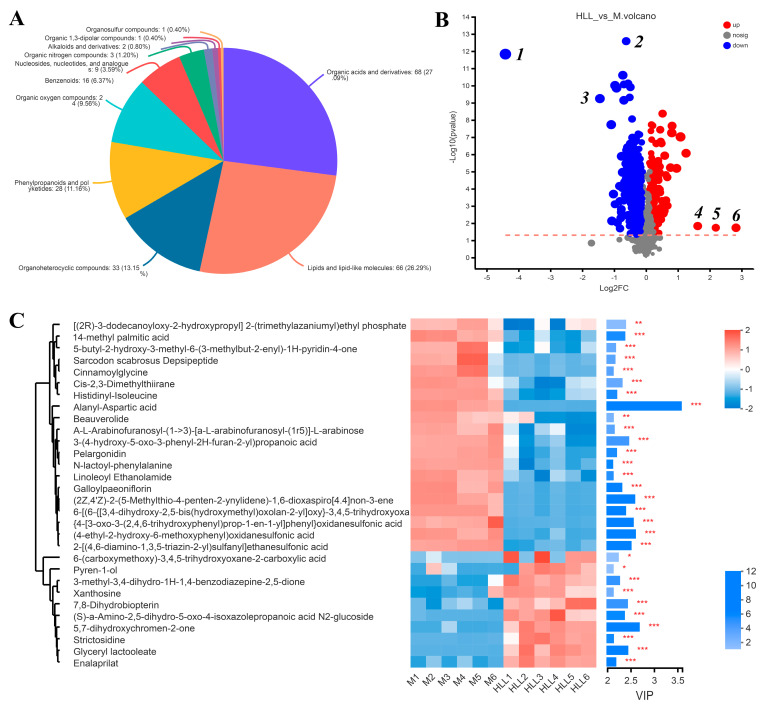
The effects of 10^2^ spores/mL *E. amstelodami* H−1 (HLL) intervention on gut metabolism in colitis mice. PCA analysis of metabolome profiles (**A**) HMDB compound classification in HLL vs. M groups. (**B**) Volcano plot analysis of the HLL vs. M groups. Red indicates the differentially upregulated metabolites, and blue indicates downregulated metabolites. The closer the points to the left and right and the upper side, the more significant the difference in expression. (**C**) Expression profile and VIP of metabolites in the HLL vs. M group. The color in the heatmap indicates the relative expression level of the metabolite. One asterisk (*) indicates *p* < 0.05, two asterisks (**) indicate *p* < 0.01, and three asterisks (***) indicate *p* < 0.001.

**Figure 7 nutrients-16-01178-f007:**
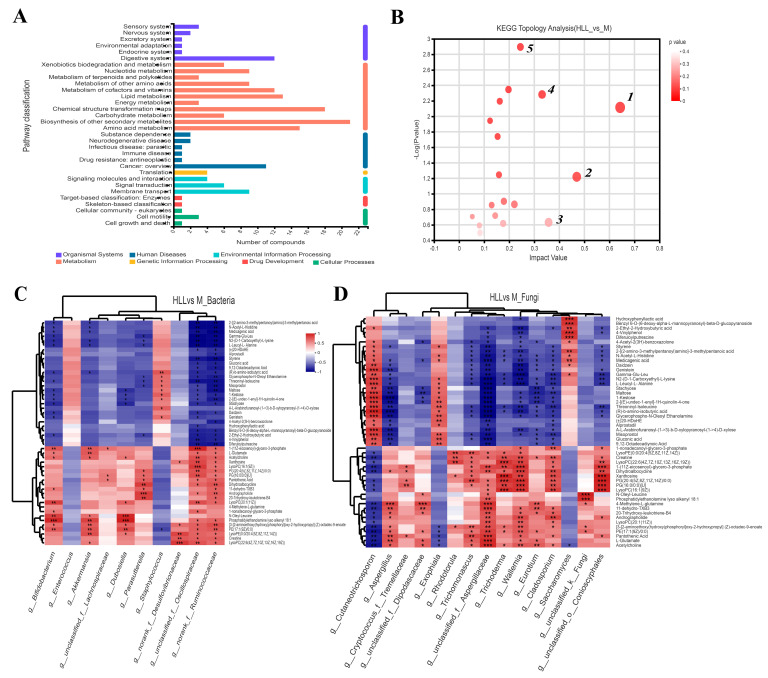
(**A**) KEGG pathway classification in HLL (10^2^ spores/mL *E. amstelodami* H−1) vs. M (model) group. (**B**) KEGG topology analysis (HLL vs. M). Each bubble represents a KEGG pathway, and the larger the bubble, the greater the importance of the pathway. Correlation between differential metabolites and (**C**) bacterial community and (**D**) fungal community in the HLL vs. M group. Differently colored grids represent the magnitude of the correlation coefficient. One asterisk (*) indicates *p* < 0.05, two asterisks (**) indicate *p* < 0.01, and three asterisks (***) indicate *p* < 0.001.

## Data Availability

Data are contained within the article and Appendix A.

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
