# Peer review of "Protective Mechanism of Eurotium amstelodami from Fuzhuan Brick Tea against Colitis and Gut-Derived Liver Injury Induced by Dextran Sulfate Sodium in C57BL/6 Mice"

_nutrients, 2024, doi:10.3390/nu16081178_

Round 1
Reviewer 1 Report
Comments and Suggestions for Authors
The manuscript is interesting. However, I have the following comments.
I: Major comments:
1. In the title the authors must include the experimental model used. Specifically mouse. This point should be considered throughout the manuscript (results and discussion).
2. The introduction is good, but I suggest briefly including mechanistic aspects that would justify the intervention and experiments carried out.
3. The methodology used is sufficient and updated.
4. The results are robust and support the discussion. But, it is necessary to improve the resolution and size of the figures. For example, it was difficult for me to identify statistically significant differences.
5. The discussion is good, but what clinical projection would the study have? One point that I suggest improving corresponds to the specific components of Eurotium amstelodami from Fuzhuan brick tea that would allow us to understand the results (anti-oxidant, anti-inflammatory effects, etc.)
II. Minor comment:
1. Improve the writing of the objective of the study.
Comments on the Quality of English LanguageThe manuscript is well written. But I suggest checking the writing.
Reviewer 2 Report
Comments and Suggestions for Authors
Congratulations on a good manuscript. The article is well-written, and well-thought-out, and the results are presented and discussed in detail. Since the recommendations regarding probiotic therapy in inflammatory bowel diseases are still ambiguous, the topic discussed here is current and fully justified.
Due to my role as a reviewer, I would like to point out some minor errors:
1. when describing figures, it is worth using full names, not abbreviations that may be incomprehensible to the person viewing them,
2. how were the animals kept after randomization? All mice from a given group in one cage? If so, how was it tested how much water with DSS did each mouse drink? How was it possible to carry out individualized measurements of occult blood in the stool? If mice lived together in one cage, one could have more severe inflammation and the other a mild one was this taken into account?
or maybe each mouse had an individual cage? but is it unethical? If so, has the impact of separation stress on intestinal inflammation been considered?
3. If faeces were collected for occult blood, were the mice allowed to eat the faeces? after all, they are coprophages.
Reviewer 3 Report
Comments and Suggestions for Authors
I'm very sorry, but the resolution of the illustrations, especially numbers 3 and 6, as well as partially others, makes them illegible, and therefore, it is impossible to assess this work. The authors should first improve the resolution of the figures.
Comments on the Quality of English Language
As it was mentioned above English is understandable and only minor corrections are required.
Round 2
Reviewer 1 Report
Comments and Suggestions for Authors
Authors answered all my comments. Therefore, the manuscript can be accepted.